# Meeting the Challenge of Eliminating Chronic Hepatitis B Infection

**DOI:** 10.3390/genes10040260

**Published:** 2019-04-01

**Authors:** Peter A. Revill, Capucine Penicaud, Christian Brechot, Fabien Zoulim

**Affiliations:** 1Victorian Infectious Diseases Reference Laboratory, Royal Melbourne Hospital at the Peter Doherty Institute for Infection and Immunity, Melbourne 3000, Australia; peter.revill@mh.org.au; 2Department of Microbiology and Immunology, University of Melbourne at the Peter Doherty Institute for Infection and Immunity, Melbourne 3000, Australia; 3Directorate, Peter Doherty Institute for Infection and Immunity, Melbourne 3000, Australia; 4University of South Florida, Tampa, FL 33612, USA; christian.brechot@pasteur.fr; 5Romark Laboratory, Tampa, FL 33607, USA; 6Global Virus Network, Baltimore, MD 21201-1009, USA; 7INSERM Unit 1052—Cancer Research Center of Lyon, 69000 Lyon, France; fabien.zoulim@inserm.fr

**Keywords:** ICE-HBV, chronic HBV disease, persistence, HBV cure, treatment, current clinical trials

## Abstract

Over 257 million people live with chronic hepatitis B virus (HBV) infection and there is no known cure. The effective preventative vaccine has no impact on existing infection. Despite the existence of drugs which efficiently suppress viral replication, treatment is usually life-long and finite therapies that cure HBV infection are urgently required. However, even if such therapies were available today, it is unlikely they would reach all of those who need it most, due to chronic hepatitis B (CHB) being largely undiagnosed across the globe and to the dire need for health systems promoting access to therapy. Considerable challenges to developing and implementing an effective HBV cure remain. Nonetheless, important advances towards a cure are being made, both in the development of a multitude of new therapeutic agents currently undergoing clinical trials, and through the establishment of a new global initiative dedicated to an HBV cure, ICE-HBV, that is working together with existing organisations to fast-track an HBV cure available to all.

## 1. Background

Today, over 1.3 billion people show evidence of past hepatitis B virus (HBV) infection, with 257 million people having progressed to chronic hepatitis B (CHB), which directly contributes to the deaths of over 880,000 people annually [1]. This public health catastrophe is made worse by the absence of effective curative interventions [2,3]. There are currently only two approaches to therapy-direct acting antivirals (DAAs) and interferon. Current DAA (nucleos(t)ide analogues (NUCs)) therapies reduce viral replication, but have minimal impact on the nuclear viral reservoir of covalently closed circular DNA (cccDNA) present in the nucleus of infected hepatocytes as a histone-bound mini-chromosome, which contributes heavily to viral persistence [2,3,4]. Also, they show no effect on integrated viral sequences which contribute to continued hepatitis B surface antigen (HBsAg) expression as well as to HBV-related carcinogenicity. Thus, they do not promote a “functional cure,” which is currently defined as hepatitis B surface antigen loss (plus or minus the production of anti-HBs) [2,3,5]. In contrast, the current “gold-standard” for immune-mediated therapy, pegylated interferon alpha (IFN-α), may result in a functional cure (hepatitis B surface antigen (HBsAg) loss) [2,5], but only in a minor proportion of recipients, with treatment response highly variable across HBV genotypes [6,7].

The inability of current therapies to induce a functional cure means that patients are invariably committed to life-long therapy. With the presence of a highly effective prophylactic vaccine, if finite curative regimens were to be developed and effectively distributed to all persons living with CHB, CHB could rapidly be eliminated as a global public health challenge. This review discusses some of the obstacles associated with meeting this admirable goal as well as describing a new collaborative approach to fast-track an HBV cure, recently instigated by the international HBV research community.

## 2. A New Global Initiative Promoting Hepatitis B Cure

The International Coalition to Eliminate Hepatitis B (ICE-HBV) (https://ice-hbv.org/) was formed in 2016 following a call to promote global collaboration in HBV cure research [2]. ICE-HBV aims to fast-track the discovery of a safe, effective, affordable and scalable cure to benefit all people living with CHB, including children and people living with hepatitis C virus (HCV), hepatitis D virus (HDV) and HIV co-infection. This will be achieved by generating knowledge, fostering collaborations and performing research to accelerate scientific innovation in collaboration with key stakeholders around the globe. ICE-HBV works with stakeholders to ensure the timely translation of discoveries into positive health outcomes and quality of life for people living with hepatitis B. Following an in-depth gap analysis, researchers leading ICE-HBV have developed a comprehensive multidisciplinary scientific roadmap for HBV cure research as a position paper for publication in early 2019. On this basis, new international collaborations have been formed in the following fields: cccDNA quantification standards, mathematical modelling of HBV cure, health systems strengthening and health policy research for HBV elimination as well as community engagement and literacy. The Coalition is now recognized as a key player in hepatitis research by the World Health Organisation (WHO) [8] and others. Next steps include the development of public–private partnerships with industry and foundations to accelerate HBV cure research, answer key research questions and fill gaps through new research projects about animal models, serum biomarkers, point-of-care assays, HBV research global investments mapping and a research protocols online repository (Figure 1). Each of these projects is designed in collaboration with key stakeholders. ICE-HBV supports the community of people living with hepatitis, particularly in recent calls from the Hepatitis B Foundation to increase HBV cure research and funding in the USA [9]. These stakeholders also include the World Hepatitis Alliance and the World Health Organisation, who collaborate with ICE-HBV to ensure that ICE-HBV’s actions align with and complement their goals to eliminate chronic HBV as a public health challenge by 2030 through universal vaccine coverage including providing a birth-dose, “finding the missing millions” of people that are undiagnosed, enabling increased access to therapy and promoting the development of finite cure regimens.

## 3. Hepatitis B Virus Life Cycle

HBV virions enter hepatocytes via the sodium-taurocholate co-transporting polypeptide (NTCP) receptor [10], where HBV virions de-envelope and the HBV DNA genome is transported to the nucleus, where it is converted to the highly stable cccDNA episome [11,12,13]. This molecule interacts with host histones to form a minichromosome, which is the transcription template for the expression of viral mRNAs, including the major transcriptional pregenomic RNA (pgRNA) template. Following encapsidation within viral nucleocapsids, pgRNA is reverse transcribed to minus then plus sense DNA, forming a relaxed circular partially double-stranded DNA molecule (RC DNA) which is then enveloped to form virions that egress from the cell. Alternatively, nucleocapsids may also be transported to the nucleus whereby RC DNA may be converted to form additional cccDNA molecules. The absence of proofreading in the viral RNA-dependent DNA polymerase (reverse transcriptase) means that errors in reverse transcription are not corrected, leading to a large quasispecies pool of HBV variants, which may in turn emerge as the dominant virus depending on selection pressures. Examples include the emergence of drug-resistant variants during lamivudine therapy [14] and precore/basal core promoter variants that emerge during hepatitis B virus e antigen seroconversion [15]. pgRNA may also undergo splicing, resulting in numerous smaller-than-genome-length molecules that are reverse-transcribed in trans and packaged into virions [16,17]. Although thought not to be critical for HBV replication, some HBV “splice variants” are associated with increased pathogenesis, including liver cancer [16,18,19,20,21].

## 4. Barriers to Hepatitis B Virus Cure

Difficulties in targeting the HBV cccDNA reservoir means that much of the current emphasis from pharmaceutical companies and academia in identifying new targets and developing CHB curative regimens lies in developing agents which target other steps of the HBV replication cycle. These include siRNA molecules which target HBV mRNAs [22,23], capsid inhibitors [24,25,26] which disrupt capsid assembly or inhibit the packaging of the viral pregenomic RNA (pgRNA) intermediate, nucleic acid polymers (NAPS) that prevent the egress of viral subviral particles [27,28] and more potent reverse transcriptase/polymerase inhibitors [29,30]. Other direct acting antiviral approaches include entry inhibitors which prevent viral entry [31] via the NTCP [10] receptor, the development of RNAse H inhibitors [32]. Approaches directly targeting HBV cccDNA, such as CRISPR-CAS9 [33,34] and TALENS [35,36] are also being tested, however, challenges associated with potential off-target effects and delivery to the infected hepatocyte must be overcome if these latter approaches are to reach the clinic. Novel direct acting antiviral approaches currently undergoing clinical trial are shown in Table 1. One of the further challenges associated with developing approaches to target cccDNA to date has been the absence of standardised methods for cccDNA detection and quantification. We applaud recent global collaborative efforts to standardise cccDNA detection and quantification through the ICE-HBV cccDNA harmonization project (Figure 1). As it is becoming increasingly difficult to obtain liver samples for analysis from HBV patients, the development of serum biomarkers which accurately represent cccDNA activity will be increasingly required.

There are also numerous immunological barriers to hepatitis B cure. These include weak innate immune responses [37,38], T-cell exhaustion [39] and defects in B cell functionality [40]. Novel immunological approaches currently undergoing clinical trial are shown in Table 1.

## 5. Clinical Advances towards Hepatitis B Virus Cure

It is likely that an eventual HBV curative regimen will require the combination of two or more of these approaches, either DAAs alone or in combination with approaches that stimulate innate and/or adaptive immune responses. For an up-to-date description of antivirals at different stages of clinical development, the reader is encouraged to regularly consult the Hepatitis B Foundation’s drug watch website [41]. A summary of current clinical trials, drawn from the Hepatitis B Foundation’s drug watch and the Clinical.trial.gov [42] websites, is presented in Table 1. These include RNAi approaches, HBV entry inhibitors, capsid inhibitors, HBsAg inhibitors and activators of innate immune responses.

In addition, Bristol Myers Squib recently concluded a Phase I/II trial on the checkpoint inhibitor Nivolumab in advanced hepatocellular carcinoma [45]. A number of these patients had chronic HBV infection. Transgene completed a Phase 1b trial on the safety and immunogenicity of their therapeutic vaccine TG1050 in nuc-suppressed CHB patients (NCT02428400; AASLD poster 426, San Francisco, 2018).

## 6. Further Challenges

Chronic hepatitis B is not a “one size fits all” disease, with striking differences in HBV natural history, disease progression and even treatment response across the globe [6,7,46,47,48]. For example, in Asia genotype C is usually associated with more progressive liver disease than genotype B, and in the western world, genotype D is usually associated with more progressive disease than genotype A. However, in Africa and India, genotype A (subtype A1) is associated with rapid progression to liver cancer in African male teenagers [49,50,51]. Indigenous Australians infected with the C4 subtype are six times more likely to have CHB than non-Indigenous Australians [52,53]. Differences in genotype also extend to treatment response, with genotype A patients responding best to IFN-α treatment and genotype D patients responding the least [6,7]. Recent in vitro studies have suggested that this may be more due to host differences rather than HBV genotype differences per se [54], with cell type rather than HBV genotype contributing most to IFN-α response. Differences have also been observed in DAA therapy, with a recent study showing that genotype A and D patients receiving the DAA tenofovir were significantly more likely to achieve a functional cure than genotype B and C patients [15,55]. Whether genotype-associated differences in treatment response are due to the virus, host, duration of infection, other factors or a combination thereof remains to be elucidated. However, it is essential that any curative regimen is pan-genotypic, to be effective in as many people living with CHB as possible.

## 7. Finding the Missing Millions

We endorse the “Find the Missing Millions” campaign recently initiated by The World Hepatitis Alliance (WHA) [56] to identify all persons living with chronic viral hepatitis, as well as the goals of the World Health Organisation to ensure that antiviral therapy reaches 50% of CHB patients worldwide by 2030. It is estimated that currently only 10% of the world’s population were diagnosed and 5% of people eligible for treatment were receiving it in 2016 [57]. Treatment rates are extraordinarily low even in western countries, reaching 8% in Australia in 2017 [58]. Of major importance to increasing diagnosis rates will be the development of point-of-care tests that enable the accurate detection of HBsAg and DNA in field applications at a limited cost. To date, two HBsAg POC test have received WHO precertification [59]. Pan-genotypic POC tests that function equally well in Africa, America, Asia, Alaska, Australia and Europe are urgently required. If a future cure is to be restricted to some stages of CHB natural history, a POC test identifying immune correlates of liver disease will also be required.

## 8. Conclusions

There has never been more interest amongst the HBV research community, public health organisations, pharmaceutical companies and the HBV-affected community in the development of an effective HBV cure. Collaborative approaches such as those fostered by ICE-HBV in partnership with the International HBV Meeting [60], WHO [61], WHA [43], The Coalition for the Eradication of Viral Hepatitis in Asia Pacific (CEVHAP) [44] and the broader HBV-affected community will be critical to achieving this goal as soon as possible.

## Figures and Tables

**Figure 1 genes-10-00260-f001:**
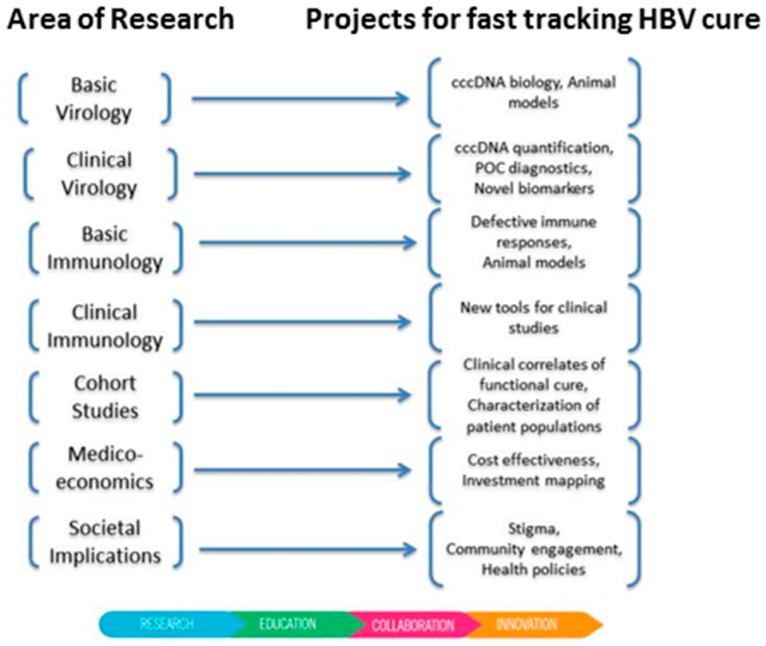
Current and future International Coalition to Eliminate Hepatitis B (ICE-HBV) projects to fast-track a hepatitis B virus (HBV) cure.

**Table 1 genes-10-00260-t001:** Current clinical trials of new hepatitis B virus (HBV) therapies (Phases I/II and II).

Approach	Name/Type	Company	Status
**Silencing HBV RNAs**	RNAi gene silencer (1.0)	Arrowhead Pharma	NCT03365947 (R)
	HBV Locked Nucleic Acid (LNA) RO7062931	Roche	NCT03038113 (R) NCT03505190 (R)
	SiRNA VIR-2218	Alnylam and Vir Biotech	NCT03672188 (R)
	Liquid nano-particle (LNP) RNAi (ARB-1462)	Arbutus Biopharma	Phase 2 (IMPACT study)
**Antisense Molecules**	IONIS-HBVRx (GSK3228836) (GSK33389404)	IONIS/GSK	NCT02981602 (R) NCT03020745 (R)
**Entry inhibitor**	Myrcludex B	Myr-pharma	NCT02888106 Recruiting hepatitis delta virus HDV/HBV coinfected patients
**Capsid Inhibitors**	GLS4	HEC Pharma	NCT03638076 (R)
	JNJ 56136379	Janssen Sciences	NCT03439488 (R) NCT03361956 (R)
	ABI-H0731	Assembly Biosciences	NCT03714152 (R)
	RO7049389	Roche	NCT 02952924 (R) NCT 03570658 (R) NCT 03717064 (A)
	AB-506	Arbutus Biopharma	Phase 1a studies completed
**HBsAg Inhibitors**	REP 2139/2165	Replicor, Canada	NCT02565719 (A) NCT02876419 (A)
**TLR7 Antagonist**	JNJ-64794964 (AL-034)	Janssen Sciences	NCT03285620 (R)
	RO7020531	Roche	NCT02956850 (R) NCT03530917 (R)
**TLR8 Antagonist**	GS-9688	Gilead, USA	NCT03615066 (R)
**Innate Immune Activators**	Inarigavir RIG-I agonist (also an HBV replication inhibitor).	Springbank Pharmaceuticals	NCT02751996 (R)
**Immune Therapy**	HBsAg monoclonal antibody	Green Cross	NCT03519113 (R)
**Therapeutic DNA Vaccine**	JNJ-64300535	Janssen Sciences	NCT03463369 (R)
**Undisclosed**	RO7239958	Roche	NCT03762681 Not yet recruiting

Sources: [43,44]. (R) = recruiting; (A) = active/recruitment closed.

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
