# Peer review of "Meeting the Challenge of Eliminating Chronic Hepatitis B Infection"

_genes, 2019, doi:10.3390/genes10040260_

Round 1

Reviewer 1 Report

This is clearly structured review or better "statement article" presenting in a concise style the major problems associated with chronic HBV infection (CHB) and the main commitments of ICE-HBV.

I just have some suggestions that may further improve clarity.

Abstract: Here I am missing a clear statement like “despite the existence of drugs efficiently suppressing viral replication / lowering viremia levels, finite therapies …”. This important point should not be neglected and I would add it on ln.16.

Ln.19: I would write… need for health systems strengthening (or promoting) access to therapy (instead of treatment response).

Ln.79-80: delete “and” at the end of the line

Pg.3; Ln.93: later it becomes clear, but at this point, when writing “finding the missing millions” I would add “of people that are undiagnosed”. Otherwise, it seems you refer to moneys.

Ln.94-95: by increased access to therapy you mostly mean current therapy (at least this is a clear, more achievable goal); a distinct goal is to promote development of finite cure regimens. I would stop the sentence here since by adding “available to all who need them” sounds to me too much as a “dream-like” aim, not realistic in reasonable time frame. I would suggest to point out more clearly the 4 different types of aims mentioned: vaccine coverage; identify undiagnosed people; increase accessibility of therapy (first that available) and last develop finite cure.

Pg.4: ln.102-103: This molecule … to form a minichromosome, which is the transcriptional template … use singular as at the beginning of sentence.

Ln.107-109: Envelopment is likely to be the main mechanism operating by HBV. Thus, I would avoid writing “either” and stop the sentence after “egress from the cell. Alternative, nucleocapsids …”.  By doing this, you tune down the 50/50 option feeling.

Ln. 113-14: add at least one ref. describing accumulation of PC/BCP variants during eAg seroconversion.

Pg.5; ln.143: statement regarding DAA and HCV: mention “including many EU states and Australia”.

Ln.151-153: For clarity I would write: These include (i) the presence of a nuclear reservoir, the cccDNA, which is not targeted by current therapies, and (ii) inadequate T and B cell responses ….

Ln.155: I would write “Furthermore,” instead of “In contrast”, since those above and the following are all gaps.

Ln.160: I would use the term strategies instead of mechanisms.

Ln.167: write “other steps” instead of “aspects”

Ln. 169-170: add one exemplary ref. of study for capsid inhibitors development/testing, as done for siRNA and for NAPS.

Ln.172: also for entry inhibitors a ref. is missing (e.g. Petersen, Nature Biotech.2008 and/or Bogomolov, J Hepatol.2016 – clinical trial, although for HDV).

Pg.6, ln.187: I think ref.43 does not fit here. You should rather use the article: Kah et al. J.Clin. Invest 2017; or Betroletti Oncoimmunology 2015.

Author Response

Reviewer 1

This is clearly structured review or better "statement article" presenting in a concise style the major problems associated with chronic HBV infection (CHB) and the main commitments of ICE-HBV.

I just have some suggestions that may further improve clarity.

Thank you, we have made all the suggested changes, as noted below.

Abstract: Here I am missing a clear statement like “despite the existence of drugs efficiently suppressing viral replication / lowering viremia levels, finite therapies …”. This important point should not be neglected and I would add it on ln.16.

Done thank you.

Ln.19: I would write… need for health systems strengthening (or promoting) access to therapy (instead of treatment response).

Done

Ln.79-80: delete “and” at the end of the line

I cannot find this error?

Pg.3; Ln.93: later it becomes clear, but at this point, when writing “finding the missing millions” I would add “of people that are undiagnosed”. Otherwise, it seems you refer to moneys.

Done

Ln.94-95: by increased access to therapy you mostly mean current therapy (at least this is a clear, more achievable goal); a distinct goal is to promote development of finite cure regimens. I would stop the sentence here since by adding “available to all who need them” sounds to me too much as a “dream-like” aim, not realistic in reasonable time frame. I would suggest to point out more clearly the 4 different types of aims mentioned: vaccine coverage; identify undiagnosed people; increase accessibility of therapy (first that available) and last develop finite cure.

Done

Pg.4: ln.102-103: This molecule … to form a minichromosome, which is the transcriptional template … use singular as at the beginning of sentence.

Done

Ln.107-109: Envelopment is likely to be the main mechanism operating by HBV. Thus, I would avoid writing “either” and stop the sentence after “egress from the cell. Alternative, nucleocapsids …”.  By doing this, you tune down the 50/50 option feeling.

Done

Ln. 113-14: add at least one ref. describing accumulation of PC/BCP variants during eAg seroconversion.

Done

Pg.5; ln.143: statement regarding DAA and HCV: mention “including many EU states and Australia”.

Done

Ln.151-153: For clarity I would write: These include (i) the presence of a nuclear reservoir, the cccDNA, which is not targeted by current therapies, and (ii) inadequate T and B cell responses ….

Done

Ln.155: I would write “Furthermore,” instead of “In contrast”, since those above and the following are all gaps.

Done

Ln.160: I would use the term strategies instead of mechanisms.

Done

Ln.167: write “other steps” instead of “aspects”

Done

Ln. 169-170: add one exemplary ref. of study for capsid inhibitors development/testing, as done for siRNA and for NAPS.

Done

Ln.172: also for entry inhibitors a ref. is missing (e.g. Petersen, Nature Biotech.2008 and/or Bogomolov, J Hepatol.2016 – clinical trial, although for HDV).

Done

Pg.6, ln.187: I think ref.43 does not fit here. You should rather use the article: Kah et al. J.Clin. Invest 2017; or Betroletti Oncoimmunology 2015.

Thank you, the correction has been made and new references included.

Reviewer 2 Report

The authors should discuss in more detail the treatment strategies currently in development for HBV. It would be nice to mention the status of all clinical trials and results as this would be very interesting to the readers. 

Author Response

Reviewer 2

Comments and Suggestions for Authors

The authors should discuss in more detail the treatment strategies currently in development for HBV. It would be nice to mention the status of all clinical trials and results as this would be very interesting to the readers.

Thank you, we have considerably updated Table 1 to reflect the status of current clinical trials.

Reviewer 3 Report

This review by Revill P et al. focusses on the ICE-HBV initiative, current challenges to finding a HBV cure and the prospective drugs against CHB that are in the pipeline. cccDNA in infected hepatocytes remains the single-most biggest challenge in finding a curative treatment for HBV and as such major global efforts are required to overcome the cccDNA problem.  Overall, the review is well-written and focuses on areas in HBV treatment and research that need a highly highly collaborative support for fruition. My comments below address issues with the figures in this review.

1) Figure 1 - It is very hard to understand the underlying message of the figure. I am not sure if the figure is needed in this review, since it is not a review solely focused on ICE-HBV initiative.Also, the ICE-HBV website is referenced in this review for readers to easily access.  If the authors feel compelled to leave this figure in the review, then I suggest that more work be put into the figure to make it more self explanatory. 

2) Figure 2 - I suggest changing this figure into a 2-column table to list the 'area of research' and the 'projects for fast tracking HBV cure'. It is not obvious why the the colored bar at the top of the figure and the pipeline arrow pointing downwards are needed. 

Author Response

Reviewer 3

This review by Revill P et al. focusses on the ICE-HBV initiative, current challenges to finding a HBV cure and the prospective drugs against CHB that are in the pipeline. cccDNA in infected hepatocytes remains the single-most biggest challenge in finding a curative treatment for HBV and as such major global efforts are required to overcome the cccDNA problem.  Overall, the review is well-written and focuses on areas in HBV treatment and research that need a highly highly collaborative support for fruition. My comments below address issues with the figures in this review.

1)      Figure 1 - It is very hard to understand the underlying message of the figure. I am not sure if the figure is needed in this review, since it is not a review solely focused on ICE-HBV initiative.Also, the ICE-HBV website is referenced in this review for readers to easily access.  If the authors feel compelled to leave this figure in the review, then I suggest that more work be put into the figure to make it more self explanatory.

We have removed Figure 1 as suggested.

2) Figure 2 - I suggest changing this figure into a 2-column table to list the 'area of research' and the 'projects for fast tracking HBV cure'. It is not obvious why the the colored bar at the top of the figure and the pipeline arrow pointing downwards are needed.

We have altered Figure 2 (now Figure 1) as suggested.

Round 2

Reviewer 3 Report

I think appropriate changes have been made to this manuscript for publication.